# Inferring spatial and signaling relationships between cells from single cell transcriptomic data

Zixuan Cang [1,3] & Qing Nie [1,2,3✉]

Single-cell RNA sequencing (scRNA-seq) provides details for individual cells; however, crucial spatial information is often lost. We present SpaOTsc, a method relying on structured optimal transport to recover spatial properties of scRNA-seq data by utilizing spatial measurements of a relatively small number of genes. A spatial metric for individual cells in scRNA-seq data is first established based on a map connecting it with the spatial measurements. The cell–cell communications are then obtained by "optimally transporting" signal senders to target signal receivers in space. Using partial information decomposition, we next compute the inter-cellular gene–gene information flow to estimate the spatial regulations between genes across cells. Four datasets are employed for cross-validation of spatial gene expression prediction and comparison to known cell–cell communications. SpaOTsc has broader applications, both in integrating non-spatial single-cell measurements with spatial data, and directly in spatial single-cell transcriptomics data to reconstruct spatial cellular dynamics in tissues.

[1] Department of Mathematics, University of California, Irvine, Irvine, CA 92697, USA. [2] Department of Developmental and Cell Biology, University of California, Irvine, Irvine, CA 92697, USA. [3] The NSF-Simons Center for Multiscale Cell Fate Research, University of California, Irvine, Irvine, CA 92697, USA. ✉email: qnie@uci.edu

Single-cell transcriptomics methods enable analyses of gene expression heterogeneities in individual cells to study cell fate decisions[1]. Dissociation of tissues into single cells allows high-throughput genomics measurements, but spatial information of cells is often lost. While single-cell transcriptomics has mainly been used to delineate cell subpopulations and their lineage relationships, recently computational tools have also been developed to infer cell–cell communications from scRNA-seq data[2,3].

For example, by comparing the average enrichment of genes involved in different cell subpopulations, one might describe the signaling activities of each subpopulation[4]. A probability model that correlates ligand and receptor (and the downstream genes) expression levels in different cells allows the inference of communications between individual cells[5]. At the level of cell populations, a similar approach based on known ligand–receptor pairs was used to derive communications between cell types[6], which can be further refined using prior knowledge of cell types[7]. Using cell type-specific enrichment of genes in known gene regulatory networks, one can also infer communication among cell clusters[8].

Despite rich details on genes contributing to cell–cell communication in scRNA-seq data, the lack of spatial information in such datasets restricts its usefulness for studying cell–cell communication in tissues with spatial structure. On the other hand, measuring gene expression in intact tissues provides spatial resolutions but the genes examined need to be selected in advance. Is it possible to better infer communications between cells located in different positions in the intact tissues using single-cell transcriptomics data with the aid of additional spatial measurements?

Several methods have been developed to pair scRNA-seq data with spatial information using spatial imaging data (e.g., in situ hybridization). For example, spatial information was obtained at cell cluster levels by identifying spatial domains with coherent gene expressions in spatial imaging data combined with scRNA-seq data[9]. At an individual cell level, similarity measurements based on correlation coefficients[10,11] or correspondence scores[12] between commonly examined genes in both spatial imaging data and scRNA-seq data were used to reconstruct spatial gene expression or map cells in scRNA-seq data to their potential spatial origins. Posterior probability estimates were carried out on spatial data described by a mixture model[13] or simplified to one-dimensional bins[14] to assign spatial origins to individual cells. Other general methods designed for the integration of multi-omics data can also be applied to integrate these two data types. Canonical correlation analysis was used to connect cells in scRNA-seq data to locations in spatial data[15], facilitating the subsequent identification of anchors across the datasets for integration[16]. Non-negative matrix factorization can also be used to construct common low-dimensional spaces of multiple datasets[17]. These methods connecting scRNA-seq data and spatial data are especially valuable for analyzing spatial patterns of different genes or cell types in embryos[10,13] and organisms with robust patterns[9,12,14].

These existing approaches focus on reconstructing spatial gene expression or estimating the spatial origins of cells in scRNA-seq data. The heterogeneity in single-cell measurements might be averaged out when determining the spatial gene expression patterns since multiple cells can be mapped to a single position. There also might be multiple highly possible spatial origins for an individual cell in scRNA-seq data. This makes existing approaches difficult to incorporate with cell–cell communications. Here we utilize the optimal transport method[18] to equip cells in scRNA-seq data with a spatial distance with single-cell resolutions by connecting with another dataset on spatial measurements of a small number of genes. Optimal transport allows natural coupling of distributions (pairing of datasets) and characterization of distances between multiple distributions (the difference between datasets or data samples represented as distributions)[18]. Recent advancements in optimal transport method development, including efficient algorithms[19], accessible library[20], and flexible formulations[21,22], enable its broader application[23–25], such as inference of developmental trajectories[26] and handling batch effects[27] in scRNA-seq data.

To connect individual cells and the spatial positions in two different measurements, we first develop a map between the two datasets by spatially optimal transporting the single cells (SpaOTsc) to spatial imaging datasets. The cell–cell distance calculated from the SpaOTsc map yields a spatial metric for scRNA-seq data. We then use this metric to establish an optimal transport plan from a probability distribution of "sender cells" to "receiver cells." Such sender cell distributions can be characterized by the expression levels of communication genes (e.g., ligands) while the receiver cells can be distinguished by the paired genes (e.g., receptors and ligand–receptor downstream genes). Our approach mimics the corresponding physical processes of ligand release by signal senders and consumption by potential receivers. After obtaining an initial cell–cell communication network, we then use a machine learning model based on ensemble of trees to estimate the spatial range of signaling for spatial cell–cell communications. While cells may communicate with each other directly through ligand–receptor interactions, a gene in one cell may affect another gene in other cells indirectly. To explore influences among genes across different cells, which may not be directly interacting in cell–cell communications, we then use partial information decomposition[28,29] to quantify the unique information provided by one gene to another gene across different cells. For a given pair of genes and a prescribed spatial distance, we quantify how likely they are interacting across different cells through space, referred to as intercellular gene–gene regulatory information flows.

We first test SpaOTsc through cross-validation of spatial gene expression predictions as well as spatial mapping of single cells with known origins in four pairs of single-cell RNA-seq and spatial gene expression measurements. We then infer cell–cell communications and intercellular gene–gene regulatory information flows for three systems.

## Results

**Overview of SpaOTsc method.** SpaOTsc method consists of two major components: (a) constructing a spatial metric for cells in scRNA-seq data and (b) reconstructing cell–cell communication networks and identifying intercellular regulatory relationships between genes (Fig. 1). A spatial metric for the cells in scRNA-seq data is first constructed using a mapping to spatial data. Using this spatial metric, we generate spatial visualization and clustering of cells and genes for scRNA-seq data. Cell–cell communication networks are then reconstructed for particular signalings. Finally, by feeding the spatial metric and scRNA-seq data to machine learning models and partial information decomposition, we infer the spatial ranges of particular signalings and quantify intercellular gene–gene regulatory information flows between genes. See more details in Methods and Supplementary Methods.

To construct a spatial metric for scRNA-seq data, we integrate it with the spatial data using optimal transport[21] (SpaOTsc). We treat the two datasets as two distributions and generate a transport cost based on the expression profile dissimilarity of shared genes across the two datasets. The dissimilarity measurements within each dataset are used to refine the mapping between these two distributions through the structured optimal transport[22]. The resulting optimal mapping depicts the probability distributions of individual cells over space.

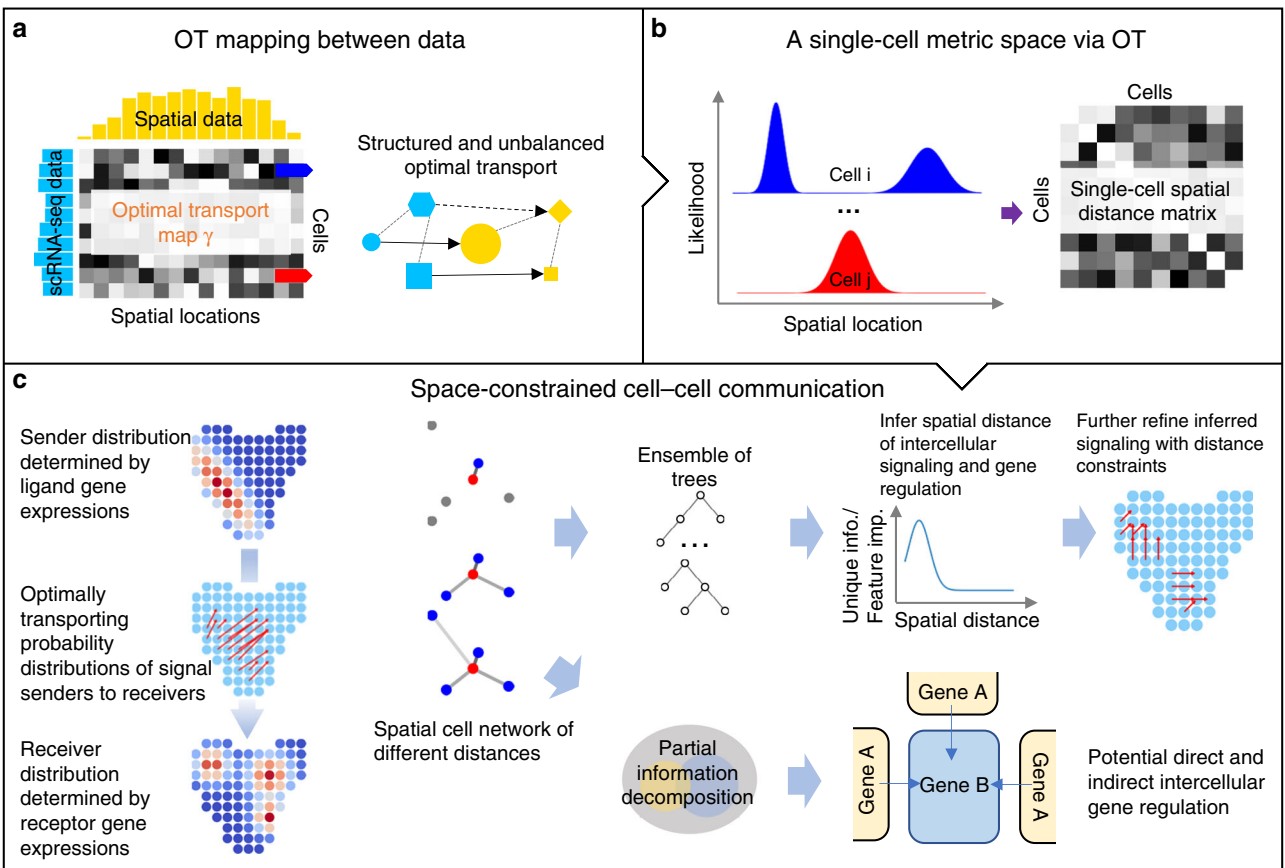

**Fig. 1 Overview of SpaOTsc. a** The unbalanced transport relaxes the mass conservation constraint (e.g. lines between circles), and the structured transport utilizes additional information (e.g. dotted links) to refine the mapping (e.g. blue hexagon). **b** Cell–cell distance is inferred by computing optimal transport distance of the spatial probability distributions of cells (rows of **γ** in **a**). **c** Calculated cell–cell distance, along with partial information decomposition and random forest models, is used to infer spatial distance of signaling and then construct space-constrained cell–cell communications and identify potential intercellular regulation between genes.

Specifically, given the spatial data ($m$ positions) and the scRNA-seq data ($n$ cells), we generate three dissimilarity/distance matrices: $\mathbf{M} \in \mathbb{R}^{n \times m}$ measuring gene expression dissimilarity between cells and positions using the common genes from the two datasets, $\mathbf{D}_{sc} \in \mathbb{R}^{n \times n}$ measuring gene expression dissimilarity among individual cells using all genes in scRNA-seq data, and $\mathbf{D}_{spa} \in \mathbb{R}^{m \times m}$ measuring the spatial distance between positions in spatial data. These matrices are fed to an unbalanced[21] and structured[22] optimal transport algorithm (Eq. (1) in Methods), which returns an optimal transport plan $\boldsymbol{\gamma}^* \in \mathbb{R}^{n \times m}$ connecting the two datasets (Fig. 1a) for the related subsequent analyses (Fig. 1b,c).

We then annotate the scRNA-seq data with a spatial metric in addition to determining a mapping between spatial positions and cells in scRNA-seq data. To this end, we infer the spatial distance between every pair of cells by computing the optimal transport distance (Eq. (2) in Methods) between their probability distributions over space (rows of $\boldsymbol{\gamma}^*$). The spatial distance among positions ($\mathbf{D}_{spa}$) is used as the transport cost. We refer to this as the cell–cell distance $\widehat{\mathbf{D}}_{sc} \in \mathbb{R}^{n \times n}$ (Fig. 1b). Additionally, the sparsity of the resulting optimal transport plan depicts the confidence of the estimated cell–cell distance.

This cell–cell distance immediately provides spatial insights when paired with conventional analysis pipelines. Visualizations on spatial arrangements of scRNA-seq can be constructed by feeding the cell–cell distance to dimension reduction methods

such as t-SNE[30] and UMAP[31,32]. Spatially localized subclusters can be classified by the cell–cell distance using clustering algorithms such as Louvain method[33]. Moreover, the genes in scRNA-seq data can be viewed as distributions on a metric space (cells equipped with the cell–cell distance). By computing the optimal transport distance between these distributions, we then derive a metric for the $n_g$ genes represented by a distance matrix $\widehat{\mathbf{D}}_g \in \mathbb{R}^{n_g \times n_g}$ assembling a gene spatial atlas.

Next, we infer cell–cell communication and intercellular gene–gene regulatory information flow over the scRNA-seq data annotated by the spatial cell–cell distance. To identify possible communications among cells mediated by ligand–receptor interactions, we formulate an optimal transport problem that transports a source probability distribution of signal sender cells to a target probability distribution of receiver cells (Eq. (4) in Methods). The expression of ligand, receptor, and downstream genes are used to estimate these sender and receiver distributions. The cell–cell distance is used as the transport cost to spatially constrain the signaling network, and the corresponding optimal transport plan $\boldsymbol{\gamma}_S^* \in \mathbb{R}^{n \times n}$ represents the likelihoods of cell–cell communications (Fig. 1c).

Knowing the spatial range of particular signaling can help further confine the inference of cell–cell communication. To infer this spatial range, we analyze a collection of trained random forest models with the downstream genes as outputs and the receptors as sample weights. The genes that highly correlate to the

downstream genes and the ligands from cells located within a spatial range are the input features. The ligand feature importance in the trained model indicates how helpful knowing the ligand expression level within the corresponding spatial range is to the prediction of downstream gene expressions. A series of spatial distances are examined, and the one with the highest ligand feature importance serves as an approximation of the spatial range for this signaling (Fig. 1c).

To interrogate whether two genes affect each other across cells through space, we utilize partial information decomposition[28,29,34] to compute the intercellular gene–gene regulatory information flow (Fig. 1c). Specifically, we estimate the unique information about a gene in a cell provided by another gene expressed in its neighboring cells within a predefined spatial distance, taking into account the information given by a collection of other genes in this cell. The gene expression in the spatial neighborhood of each cell is estimated by a weighted average based on the spatial metric of cells in scRNA-seq data. Both cellular gene expression and spatial neighborhood gene expression are summarized into histograms using Bayesian blocks[35]. These histograms are fed to discrete partial information decomposition algorithms[28,34] (Eq. (3) in Methods). By iterating over different spatial distances, this approach yields a directed network of genes annotating possible interactions between genes across cells under different spatial distances.

**Accuracy of SpaOTsc mapping and comparison to other methods.** The mapping between scRNA-seq data and spatial data obtained by SpaOTsc is the foundation of the subsequent analyses. To evaluate this mapping, we utilized four scRNA-seq datasets paired with spatial data from zebrafish embryo, *Drosophila* embryo, and mouse visual cortex. Two different scRNA-seq datasets on measurements of 6hpf zebrafish embryo were used. The first dataset[13] has 851 cells and 10495 genes, and it has been previously used for analyzing spatial data[13]. The second dataset[36] has 5693 cells and 30677 genes, and it can be used to test our method in handling unbalanced datasets since the number of cells in scRNA-seq data is ~90 folds more than positions in spatial data. The spatial reference dataset[13] consisting of 64 spatial positions and 47 genes was used for both single-cell datasets. For the *Drosophila* embryo, the scRNA-seq data[10] has 1297 cells and 8925 genes, and the spatial data[10] has 3039 spatial positions and 84 genes. The mouse visual cortex scRNA-seq dataset[37] has 15413 cells and 45768 genes, and the corresponding spatial dataset[38] has 1549 spatial positions and 1020 genes. The details on data acquisition and preprocessing can be found in Datasets and processing in Methods.

For the zebrafish embryo and *Drosophila* embryo, we carried out leave-one-out cross-validation of spatial expression prediction for each gene in spatial data using the scRNA-seq data. When predicting the spatial expression for each gene, we excluded the gene for prediction in the spatial data. The quality of the reconstructed spatial gene expressions was evaluated by Spearman's correlation coefficient, the area under the receiver operating characteristic curve (AUC), and root-mean-square error (RMSE). When comparing to binary spatial data, AUC is used to evaluate this classification problem. RMSE is used when the spatial data is continuous. Three other established methods for spatial gene expression prediction DistMap[10], Achim, et al.[12], and Seurat v1[13] were used for comparison. All three methods provide mapping matrices between scRNA-seq data and spatial data, which were used to reconstruct spatial gene expression via a weighted average. Our method has shown high accuracy in the three pairs of datasets tested (Fig. 2a-d, Supplementary Figs. 1–4), achieving an AUC of 0.88 in *Drosophila* dataset and 0.95/0.94 for

the first/second pairs of zebrafish datasets (Table 1). The performance on the second much larger scRNA-seq dataset of zebrafish embryo is only slightly inferior to that on the first smaller dataset (Table 1). This indicates the capability of SpaOTsc at handling unbalanced data size while combining scRNA-seq data and spatial data. SpaOTsc exhibits a more noticeable improvement in terms of evaluation metrics upon other methods on the *Drosophila* embryo datasets, which contain more detailed spatial data compared to the zebrafish embryo datasets (Table 1). This implies that our method is potentially more robust and effective for spatial data with higher spatial resolution. We also investigated the prediction accuracy of our method using three other data normalization procedures, and found consistent results (Supplementary Fig. 5). The observed robustness in the prediction under different preprocessing procedure is partly due to the usage of ranking based correlation coefficients for similarity measurements.

In the original mouse visual cortex datasets, the cells were annotated with their original layer in the intact tissue. The problem of predicting the original spatial region for scRNA-seq data is used to evaluate our method in a multiclass classification problem (Fig. 2e,f). A micro $F_1$ score (harmonic mean of precision and recall) of 0.48 was achieved (compared to 0.09 for a baseline model that always predicts the label of the majority population). We also evaluated the importance of using unbalanced optimal transport and the benefit of incorporating the information of all genes in scRNA-seq data through structured optimal transport. This was done by altering the parameters on the weights for the unbalanced and structure terms (see Eq. (1) in Methods). Both unbalanced and structured optimal transport yield improved performance upon balanced and unstructured configurations in all tested cases (Supplementary Fig. 1).

**Space-constrained visualization and clustering.** We applied SpaOTsc to analyze the spatial aspect of scRNA-seq datasets. Visualizations of scRNA-seq data with spatial details were produced by feeding the cell–cell distance to commonly used dimension reduction algorithms (Fig. 3a,b, Supplementary Figs. 6–11). Gene–gene "distances" that are used to depict the difference in spatial expression patterns can also be generated using the cell–cell distance. Such gene–gene difference provides a gene spatial atlas, in which networks of genes with edges indicating similarity in spatial expression patterns are obtained when the scRNA-seq data is equipped with a cell–cell spatial distance (Fig. 3c,d, Supplementary Figs. 12, 13).

For the mouse visual cortex, a spatial axis from layer L2/3 through L6 was reconstructed for the scRNA-seq data (Fig. 3a). The spatial visualization of scRNA-seq data shows a consistent spatial colocalization of different cell types. For example, somatostatin (Sst) expressing cells are relatively abundant across space and are colocalized with vasoactive intestinal peptide (Vip) and Parvalbumin (Pvalb) cells (Fig. 3a). Direct interactions between Sst and Pvalb neurons, and Vip and Sst neurons are known to regulate neuron activity[37,39]. The spatial visualization suggests that Sst neurons are preferentially placed in the middle of Vip and Pvalb neurons in space, indicating the indirect interaction between Vip and Pvalb neurons through Sst neurons. In contrast, the low-dimensional visualization based only on scRNA-seq data does not show such spatial arrangements (Supplementary Fig. 11).

For the *Drosophila* embryo, the spatial visualization of scRNA-seq data successfully reconstructs the dorsal-ventral and posterior-anterior axes (Fig. 3b). Spatially localized subclusters of the same cell type were also identified, further revealing the relationship between cell heterogeneity and spatial arrangement.

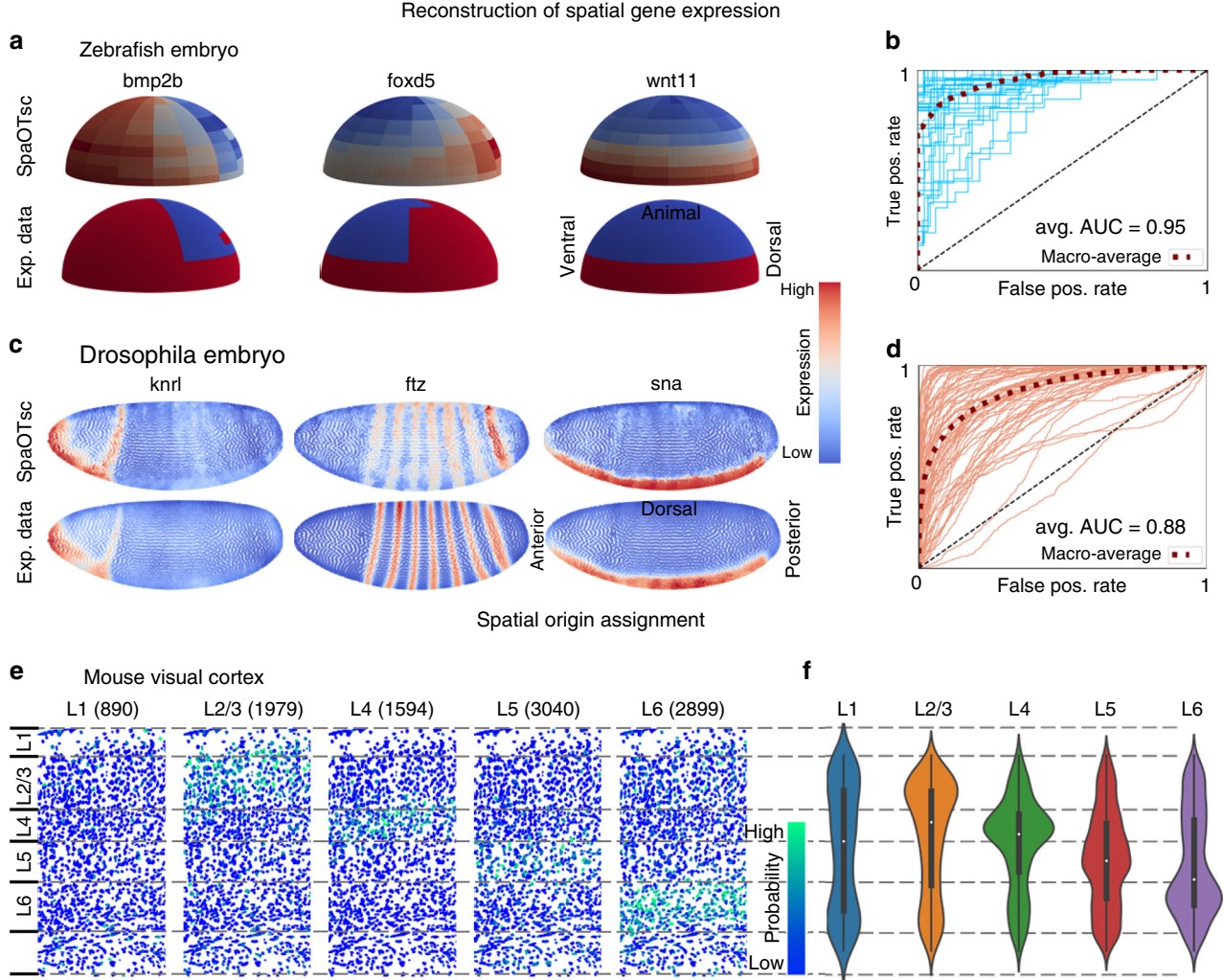

**Fig. 2 Validation of SpaOTsc using three systems. a** Predicted spatial expressions for the zebrafish embryo (both data from ref. [13]). **b** The receiver operating characteristics (ROC) curves of leave-one-out cross-validation (LOO CV) of the spatial expression prediction for the zebrafish embryo data. **c** Predicted spatial expressions for the *Drosophila* embryo (both data from ref. [10]). **d** The ROC curves of LOO CV of the spatial expression prediction for the *Drosophila* embryo spatial data. **e** Assignment of spatial positions to the scRNA-seq data for the mouse visual cortex (spatial data from ref. [38]; scRNA-seq data from ref. [37]). Each column depicts all cells from the spatial data in the visual cortex. For example, in column one, the color of cells represents the average probability of the spatial origin of the 890 cells in scRNA-seq data labeled with spatial origin L1. **f** Violin plots along L1-L6 axis of the mapped spatial origins for single cells from each subregion. Inside the violin plots are standard boxplots (median, 25th perceltile, 75th percentile, the bigger of minimum value and 25th percentile – 1.5 interquartile range, and smaller of maximum value and 75th percentile + 1.5 interquartile range). The numbers of data points for the violin plots from left to right are 890, 1979, 1594, 3040, 2899, respectively.

**Table 1 Performance comparison by leave-one-out cross-validation of spatial gene expression prediction.**

|  | SpaOTsc | DistMap | Achim | Seurat (v1) |
|---|---|---|---|---|
| D. Em. AUC (b.) | 0.876 | 0.818 | 0.847 | – |
| D. Em. $R_S$ (b.) | 0.495 | 0.409 | 0.451 | – |
| D. Em. RMSE (c.) | 0.225 | 0.303 | 0.278 | – |
| D. Em. $R_S$ (c.) | 0.424 | 0.339 | 0.379 | – |
| Z. Em. 1 AUC (b.) | 0.952 | 0.939 | 0.929 | 0.942 |
| Z. Em. 1 $R_S$ (b.) | 0.681 | 0.663 | 0.645 | 0.667 |
| Z. Em. 2 AUC (b.) | 0.936 | 0.926 | 0.887 | 0.911 |
| Z. Em. 2 $R_S$ (b.) | 0.657 | 0.642 | 0.579 | 0.619 |

For each gene in spatial data, the other genes in spatial data and the entire scRNA-seq data are used to predict the spatial expression of this gene. The accuracy of gene expression prediction is evaluated by Spearman's correlation to continuous (c.) and binarized (b.) spatial expression data. A root-mean-square-error is used to assess the difference between prediction and ground truth that are both linearly mapped to an interval from 0 to 1. The area under the receiver operating characteristics curve (AUC) is used to assess the accuracy by considering the problem as a binary (on/off) classification. Three other recognized methods are evaluated which are DistMap (ref. [10]), Achim (ref. [12]), and Seurat v1 (ref. [13]). Both zebrafish examples use the spatial data from ref. [13]. The two zebrafish examples use scRNA-seq data from ref. [13] (Z. Em. 1) and ref. [36] (Z. Em. 2), respectively. The *Drosophila* example (D. Em.) uses data both from ref. [10]. The application of Seurat (v1) to the *Drosophila* dataset failed.

Spatial visualization of scRNA-seq data based on cell-cell distance

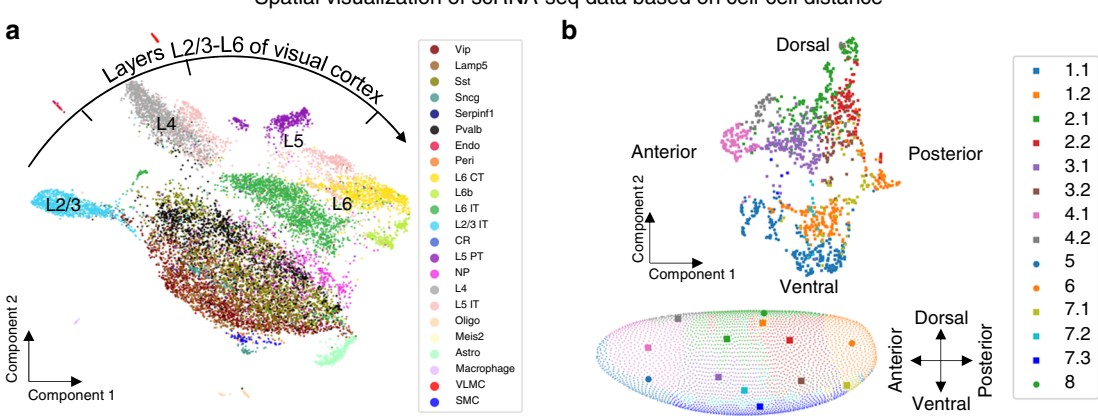

Visualization of genes based on their spatial expression similarity (gene spatial atlas)

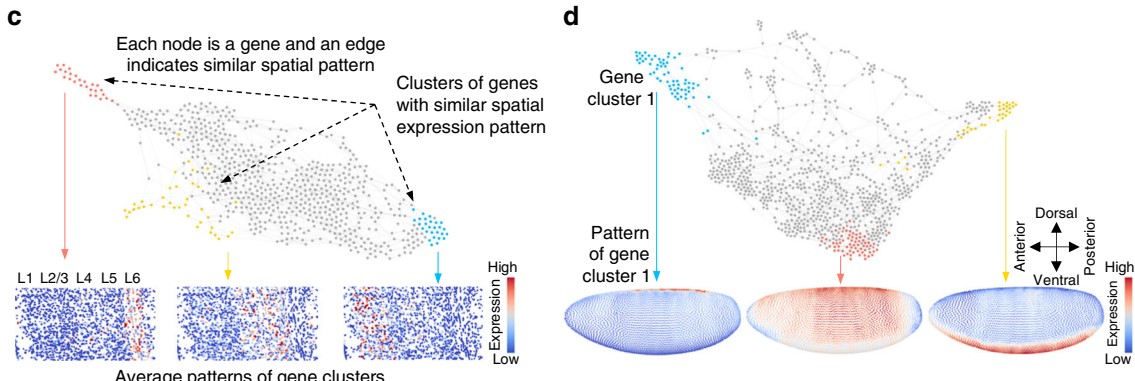

**Fig. 3 Metric spaces and spatial gene atlases for scRNA-seq data. a** A low-dimensional spatial visualization (UMAP) of mouse visual cortex scRNA-seq data (ref. [37]) using the cell–cell distance inferred by SpaOTsc with the spatial data (ref. [38]). The cell labels are taken from ref. [37]. **b** Similar to (a) but for *Drosophila* embryo data (ref. [10]). **c, d** The gene spatial atlases for mouse visual cortex data (**c**) and *Drosophila* embryo data (**d**) consisted of collections of highly variable genes where nodes represent genes and edges indicate similarity in spatial pattern.

Gene spatial atlases were constructed to classify spatial gene expression patterns (Fig. 3c,d). For the mouse visual cortex, we identified genes that are enriched in certain spatial regions such as the clusters of genes expressed at L2/3, L4-L5, and L6 (Fig. 3c) as well as genes that have no apparent spatial localization behavior (Supplementary Fig. 13). For the *Drosophila* embryo, we identified gene clusters that are highly expressed in the dorsal or ventral side, and a gene cluster that exhibits a smooth dorsal-to-ventral gradient which may contribute to dorsal-ventral patterning (Fig. 3d).

**Reconstruction of cell–cell communication in space.** We first constructed a cell–cell distance for the scRNA-seq data of zebrafish embryo[13,36] at 6hpf using the spatial data[13] of the same developmental stage. We inferred cell–cell communication in scRNA-seq data through Wnt signaling and bone morphogenetic protein (BMP) signaling (Fig. 4a,b) using known ligand, receptor, and downstream genes. The cell–cell communication was mapped to space by constructing a position to position communication flow using the SpaOTsc mapping matrix between scRNA-seq data and spatial data. Based on the position to position communication, we approximated the direction of signal flow from the signal sending positions (Fig. 4a). The cell–cell communication was also summarized into a cluster–cluster communication matrix revealing the communication between cell types (Fig. 4b). The genes used in signaling analysis are listed in Section 3 of Supplementary Methods.

We found significant Wnt signaling, an important development regulator, from the ventral side to the dorsal side along the margin, which may contribute to axis specification[40]. Most Wnt signaling activity was identified to take place within the mesoderm. A significant group of Wnt ligand sending cells was identified at the ventrolateral margin (depicted by arrow length in Fig. 4a) indicating the regulation of the later formation of posterior mesoderm through Wnt signaling[41]. Interestingly, a subgroup of ectodermal cells was found near the dorsal margin sending signals to a subgroup of mesodermal cells (cluster 10 to cluster 9 in Fig. 4b).

Significant BMP signaling, an essential regulator on development growth, was identified at the ventral side, which is consistent with the established BMP signaling gradient along the ventral-dorsal axis[42]. While Wnt signaling was mainly identified in the mesoderm, BMP signaling was found to be enriched across endoderm, mesoderm, and ectoderm at the ventral side. Furthermore, we found a secondary hotspot of BMP signaling receivers colocalized with Wnt signaling receivers at the dorsal side (cluster 9 in Fig. 4b) which supports the suggested interaction between Wnt and BMP signaling in early embryo development[43]. This subgroup of both Wnt and BMP receivers is located in the mesoderm, indicating possible crosstalks between Wnt and BMP signaling through the mesodermal layer.

Next we performed a similar analysis for fibroblast growth factor (FGF) signaling (Supplementary Fig. 14). While the identified BMP signaling activity was found to be strong on the ventral side, the inferred FGF signaling was found to be more

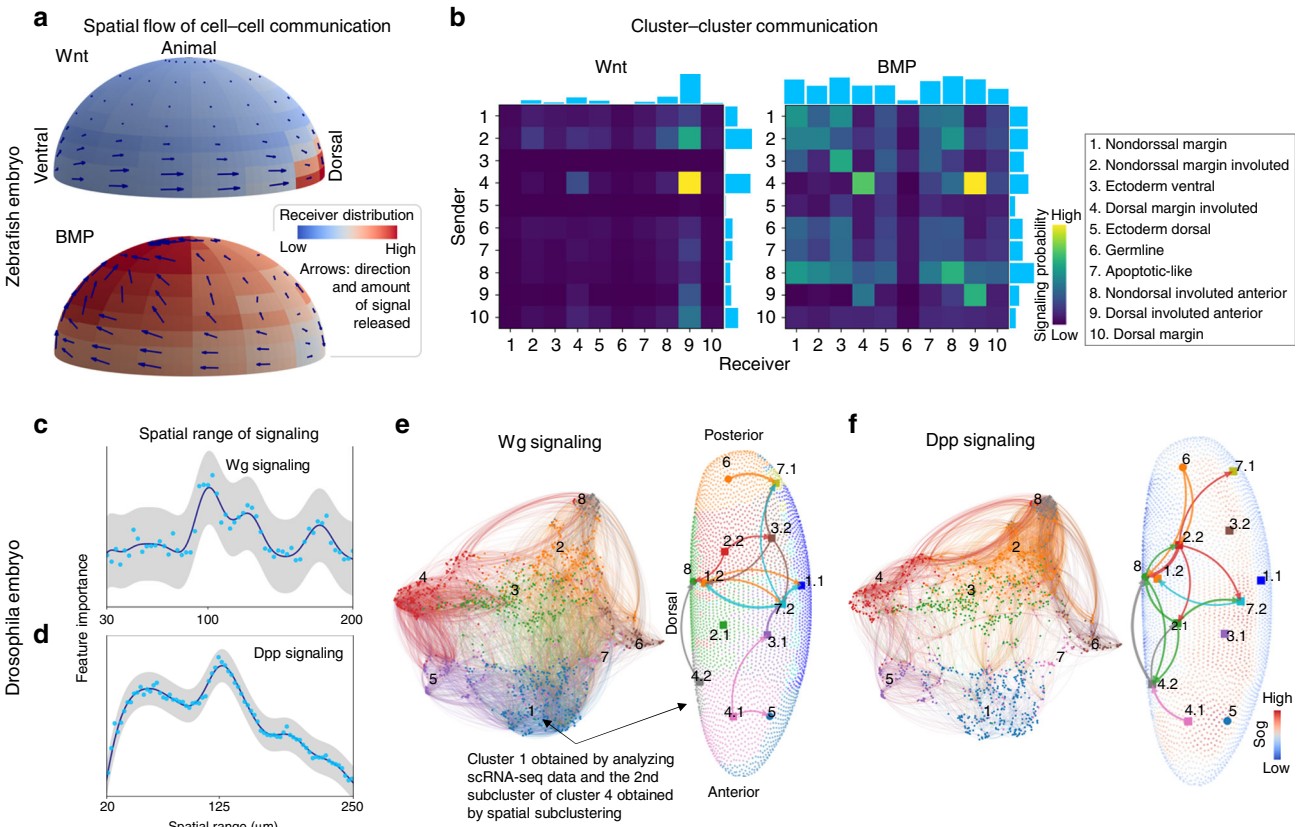

**Fig. 4 Reconstruction of cell–cell communications in space. a** (zebrafish embryo) Wnt and BMP signalings interpolated from the SpaOTsc cell–cell communication matrix and mapped to space using the mapping between cells and positions. The arrow lengths indicate the signal sending probability of the position and the color shows the estimated signal receiver probability distribution over space. The scRNA-seq data from ref. [36] and spatial data from ref. [13] were used. **b** (zebrafish embryo) Wnt and BMP signaling summarized into cell clusters. **c, d** (*Drosophila* embryo) Spatial ranges of Wg and Dpp signaling inferred using a series of sets of random forest models. The gray band represents the 95% confidence interval. The experiment was repeated three time with similar results. **e** (*Drosophila* embryo) Left: cell–cell communications of Wg signaling at the single-cell level using a visualization (UMAP) constrained by cell–cell distance. The color of the link is marked by the color of the sending cells, based on the clustering using only scRNA-seq data. Right: cluster–cluster communications of Wg signaling based on SpaOTsc spatial subclustering (subcluster the previously determined clusters based only on scRNA-seq data using the cell–cell distance). **f** (*Drosophila* embryo) Dpp signaling in space plotted similar to (**e**).

active on the dorsal side. This observation is consistent with a prior study, suggesting a down-regulation mechanism on BMP by FGF signaling[44].

For the *Drosophila* embryo[10], we used SpaOTsc to analyze cell–cell communications with a focus on wingless (Wg) and decapentaplegic (Dpp) signaling (Fig. 4c–f). To fully utilize the fine resolution of this spatial data, we first estimated the spatial ranges of the signalings to restrict the cell–cell communication networks in space.

Wg, an invertebrate analog of Wnt that plays an essential role in growth, polarity and patterning, was previously shown to act in a range of 50–100 µm[45]. The spatial range of Wg signaling inferred using SpaOTsc was about 100 µm (Fig. 4c, Supplementary Fig. 15). After estimating the probability of signaling between each pair of cells constrained by the spatial distance, the cell–cell communications could be summarized into cell subclusters (Fig. 4e). Interestingly, a thin strip of cells located near the lateral-ventral part of the embryo was found to be both sources and targets of Wg signaling. Moreover, Wg signaling was abundant at the lateral side of the embryo with the direction biased toward the posterior. This finding explains a previous observation that Wg signaling is crucial to the growth of the posterior[46], and further predicts a subpopulation of cells at the posterior-ventral domain that receives Wg signaling from their

neighbors. In addition, one can prioritize the most significant cell–cell connections by adjusting a scaling parameter (η in Supplementary Methods Eq. 34), which determines whether a gene is sufficiently expressed to be included in the cell–cell communication analysis (Supplementary Fig. 16).

Dpp, the *Drosophila* homolog of BMP which is an essential morphogen regulating the patterning in early *Drosophila* embryo development, was found to have a longer signaling spatial range of 125 µm (Fig. 4d, Supplementary Fig. 17). The most active Dpp signaling was predicted to occur at the lateral side where short gastrulation (Sog) expression is predicted to be abundant, supporting a prior result that Dpp signaling undergoes a long-range transport facilitated by Sog during dorsal-ventral patterning[47] (Fig. 4f). Interestingly, the strong Wg source located near the ventral side was also identified as a significant target of Dpp signals from the dorsal side. When we compared Wg and Dpp based cell–cell communications inferred by SpaOTsc with another inference method[5] which does not include spatial information, we found that SpaOTsc makes predictions that are more biologically feasible and more consistent with the prior knowledge (Supplementary Figs. 18–22).

Epidermal growth factor (EGF) signaling, another key regulator of dorsal-ventral patterning[48] was also inferred (Supplementary Fig. 23). Similar to Dpp that regulates dorsal-ventral patterning,

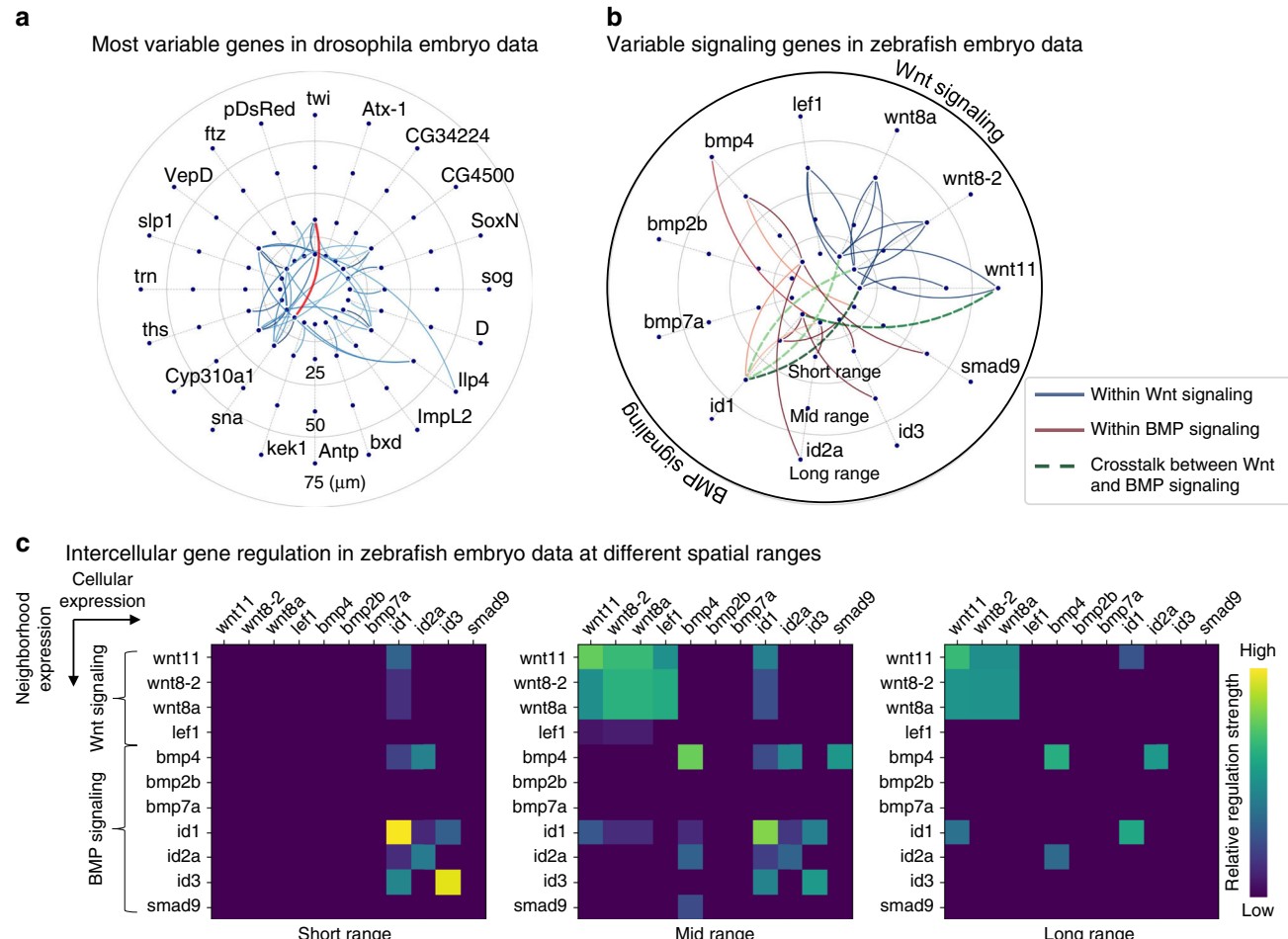

**Fig. 5 Intercellular gene–gene regulatory information flows. a** (*Drosophila* embryo) The intercellular gene–gene regulatory information flow for the top 20 variable genes in *Drosophila* embryo scRNA-seq data. For example, gene Twist in the 25 μm shell is connected with gene Snail (red curve), suggesting Snail is directly or indirectly affected by Twist in neighbor cells within a spatial distance of 25 μm. **b** (zebrafish embryo) The intercellular gene–gene regulatory information flow for the variable genes involved in Wnt signaling or BMP signaling. Relative distances are considered where short, medium and long range corresponds to 1/8, 1/4 and 1/2 of the embryo radius. **c** Heatmaps of the information flows at different spatial scales showing the intercellular regulation within and across the two signaling modules.

EGF signaling was found to be strong along the dorsal-ventral axis. The inferred EGF signaling is more active in the posterior in contrast to Dpp signaling that is stronger in the anterior.

**Identification of intercellular gene–gene information flows.** In the previous section, we inferred relationships between cells (the cell–cell communication network) based on known genes involved in signaling. Here we attempt to identify the spatial influence of one gene on another gene by computing the intercellular gene–gene regulatory information flow.

For the drosophila embryo, we inferred such a flow for a set of most variable genes under different spatial ranges to predict which gene in one cell may affect another gene in a different cell located within an estimated maximal distance (Fig. 5a). For example, gene Twist is connected to gene Snail at a spatial distance of 25 μm (red curve in Fig. 5a), suggesting that Snail is directly or indirectly affected by Twist in neighbor cells within the spatial distance. These two genes are known to be important during mesoderm formation[49].

For the zebrafish embryo, we analyzed genes that may have links between Wnt and BMP to study crosstalk between these two signalings (Fig. 5b, c). A subset of variable genes was used to infer the information flow, confirming the intercellular regulatory

relationships within Wnt signaling or BMP signaling (solid curves in Fig. 5b). We also found several significant connections between genes from Wnt signaling and BMP signaling (dashed curves in Fig. 5b), suggesting potential interactions between Wnt and BMP signaling. Moreover, significant connections between a downstream gene of BMP signaling *id1* (inhibitor of DNA binding 1) and the Wnt ligand genes were identified. This finding is consistent with a previous suggestion that *id1* is a mediator for the crosstalk between Wnt signaling and BMP signaling[50].

To investigate whether the number of background genes affects the inference of gene–gene regulatory information flows, we systematically increased the number of background genes from 1 gene to 300 genes for the background genes. Consistent results were obtained once more than 50 genes were used as the background genes for the inference (Supplementary Figs. 24–29).

**Applications to spatial transcriptomics datasets.** To investigate if SpaOTsc is applicable to the inference of cell–cell communications using spatial transcriptomics data, we used two different datasets for mouse olfactory bulb: a Slide-seq dataset[51] containing 26316 cells with 18838 genes and an RNA seqFISH+ dataset[52] containing 2050 cells with 10000 genes. In addition, we utilized one scRNA-seq dataset[53] containing 51426 cells with 18560

genes. The scRNA-seq dataset consists of six samples from three physiological conditions: wild type (WT), olfactory trained (TR), and naris occluded (OC). By selecting secreted ligands in a database of more than a thousand ligand–receptor pairs[3], we identified a list of 1157, 989, and 758 pairs in the scRNA-seq, Slide-seq, and RNA seqFISH+ datasets, respectively.

We first inferred cell–cell communications in the spatial transcriptomics datasets without using the scRNA-seq dataset. A spatial transcriptomics dataset annotated with spatial distances between cells directly computed from the spatial coordinates in the data does not require the usage of the first part of SpaOTsc, the part to integrate spatial data and scRNA-seq data, denoted as SpaOTsc-integration (Fig. 1a,b). The second part of our method, denoted as SpaOTsc-communications (Fig. 1c), was used to analyze the spatial transcriptomics datasets. We found in both spatial datasets that the signal sender cells exhibit more spatial localization pattern, and the signal receivers are more scattered over the space (Fig. 6a, b, Supplementary Figs. 30, 31). For example, a strip of cells in the middle of the Slide-seq data (Fig. 6b) and the top portion of the RNA seqFISH+ data (Fig. 6a) are the signal senders. Individual ligands such as *Apoe* and *Ptn* are abundant across the whole sample and *Trf* is sparse and located in the left and right side of the domain (Fig. 6c). The intercellular gene–gene regulatory information flow for the top variable genes in the Slide-seq dataset shows abundant connections for the gene *Pcp4* (Fig. 6d), a gene suggested to be expressed in neuronal origins[54].

We next integrated the scRNA-seq dataset with the two spatial transcriptomics datasets, respectively, using SpaOTsc-integration. As a result, cells in scRNA-seq data were mapped into space after using SpaOTsc-communications (Fig. 6a,b, Supplementary Figs. 32, 33). To study the similarities between the three different physiological conditions, only available in the scRNA-seq data, we carried out a clustering on the scRNA-seq data with all cells from different conditions and samples (Supplementary Fig. 34) to compare the average cell–cell communications between different clusters. Overall the six samples have a similar cell–cell communication profile, however the OC samples are a bit more different from the TR and WT samples (Fig. 6e). A similar result was obtained when integrating with the Slide-seq dataset (Supplementary Fig. 35).

## Discussion

Overall, we have shown the capabilities of SpaOTsc to (1) map between scRNA-seq data and spatial data, (2) infer spatial distances between single cells, (3) quantitatively compare spatial gene expression patterns, (4) reconstruct spatial cell–cell communications, (5) estimate the spatial range of particular types of intercellular signaling, and (6) identify gene pairs that potentially intercellularly regulate each other.

The mapping accuracy of SpaOTsc has been demonstrated by gene expression reconstruction validation on zebrafish embryo and *Drosophila* embryo datasets, along with spatial origin assignments to the scRNA-seq data of mouse visual cortex. Unlike previous mapping methods, the mapping of a cell–position pair depends on not only the gene expression profile similarity between this pair but also the mapping of all other pairs. The structured nature of our optimal transport method allows us to fully utilize the scRNA-seq data, which is especially useful when the spatial data only partially represents the cell types in scRNA-seq data.

The spatial metric for cells in scRNA-seq obtained using SpaOTsc allows one to carry out spatial analyses of all genes at a single-cell resolution. Inferring the spatial distance between two cells by comparing their estimated spatial probability distributions provides a useful coupling between these two cells, quantifying the confidence of the estimated cell–cell distance. In addition, this spatial metric annotated scRNA-seq data can be fed

to different spatial transcriptomics analysis pipelines such as Giotto[55]. Beyond application to data analysis, the spatial metric for scRNA-seq data can also be used for modeling approaches. For example, ordinary (or partial) differential equations on graphs might be introduced using this metric to study the dynamics of intracellular and intercellular gene regulation.

Computationally, the cell–cell distance inference requires an iterative calculation of optimal transport over all pairs of cells. Although effective approximation was made by only using a small number of landmark positions, the computation can become intractable when the dataset is excessively large. Improvement can be made by first constructing a graph partially representing the distances between cells, and approximating the full cell–cell distance matrix using the methods to estimate pairwise distances designed for large graphs[56].

Adding spatial constraints in cell–cell communication inference is critical to spatial analysis of gene–gene regulations across cells. However, our approach does not consider the time delay that may take place in cell–cell communication. Such delay may include the diffusion time of ligand or the reacting time of the intracellular cascades. It is potentially beneficial to include this effect in studying spatially regulated cell–cell communication, and dynamical systems models or more sophisticated probability models might be needed for more accurate inference.

Other than inferring cell–cell communications based on known genes involved in specific signaling, the estimation of the spatial range of signalings and identification of new gene pairs that might affect each others' expression across cells are potentially instrumental in spatial analysis of gene expression data. Further incorporation of gene–gene regulatory networks in our spatial analysis tools can be very fruitful in studying spatial gene regulations. Finally, SpaOTsc is generally applicable to datasets where reasonable similarity measurement between single cells and spatial positions are obtainable. Since single-cell spatial transcriptomics data[52,57] naturally resembles a (spatial) metric space of a collection of individual cells, SpaOTsc is also directly applicable to the high-throughput spatial data. The SpaOTsc-integration utility provides a useful tool for the integration of scRNA-seq data with the spatial transcriptomics data to fully utilize more easily available scRNA-seq datasets under various biological conditions.

## Methods

Full details of the theoretical background and implementation of SpaOTsc can be found in Supplementary Methods.

**SpaOTsc model**. SpaOTsc constructs a mapping between the $n$ cells in scRNA-seq data and the $m$ positions in spatial data by solving an optimal transport problem[20] given three dissimilarity/distance matrices, $\mathbf{M} \in \mathbb{R}^{n \times m}$ for the gene expression dissimilarity between cells and locations, $\mathbf{D}_{sc} \in \mathbb{R}^{n \times n}$ for the gene expression dissimilarity among cells, and $\mathbf{D}_{spa} \in \mathbb{R}^{m \times m}$ for the distances among spatial locations. The optimal transport plan $\boldsymbol{\gamma}^*$ is obtained by solving

$$\underset{\boldsymbol{\gamma} \in \mathbb{R}_+^{n \times m}}{\arg\min} \Bigg[ (1-\alpha) < \boldsymbol{\gamma}, \mathbf{M} >_F \\ + \rho \big( \mathrm{KL}(\boldsymbol{\gamma} \mathbf{1}^m | \boldsymbol{\omega}_1) + \mathrm{KL}(\boldsymbol{\gamma}^T \mathbf{1}^n | \boldsymbol{\omega}_2) \big) \\ + \alpha \sum_{i,j,k,l} L(\mathbf{D}_{sc}(i,k), \mathbf{D}_{spa}(j,l)) \boldsymbol{\gamma}_{i,j} \boldsymbol{\gamma}_{k,l} \Bigg] \quad (1)$$

where $\boldsymbol{\omega}_1, \boldsymbol{\omega}_2$ are weight vectors and $L$ measures the difference between scaled dissimilarities/distances. The first term quantifies the major transport cost, the second penalty term promotes weight conservation (unbalanced transport)[21], and the last term preserves the distance within datasets through the mapping (structured transport)[22]. The spatial cell–cell distance $\widehat{\mathbf{D}}_{sc}$ is then computed based on $\boldsymbol{\gamma}^*$ using the optimal transport distance:

$$\widehat{\mathbf{D}}_{sc}(i,j) = \min_{\boldsymbol{\gamma} \in \Gamma} < \boldsymbol{\gamma}, \mathbf{D}_{spa} >_F,$$
$$\Gamma = \{ \boldsymbol{\gamma} \in \mathbb{R}_+^{m \times m} : \boldsymbol{\gamma} \mathbf{1}^m = \boldsymbol{\gamma}_i^* / \sum_k \boldsymbol{\gamma}_{i,k}^*, \boldsymbol{\gamma}^T \mathbf{1}^m = \boldsymbol{\gamma}_j^* / \sum_k \boldsymbol{\gamma}_{j,k}^* \}. \quad (2)$$

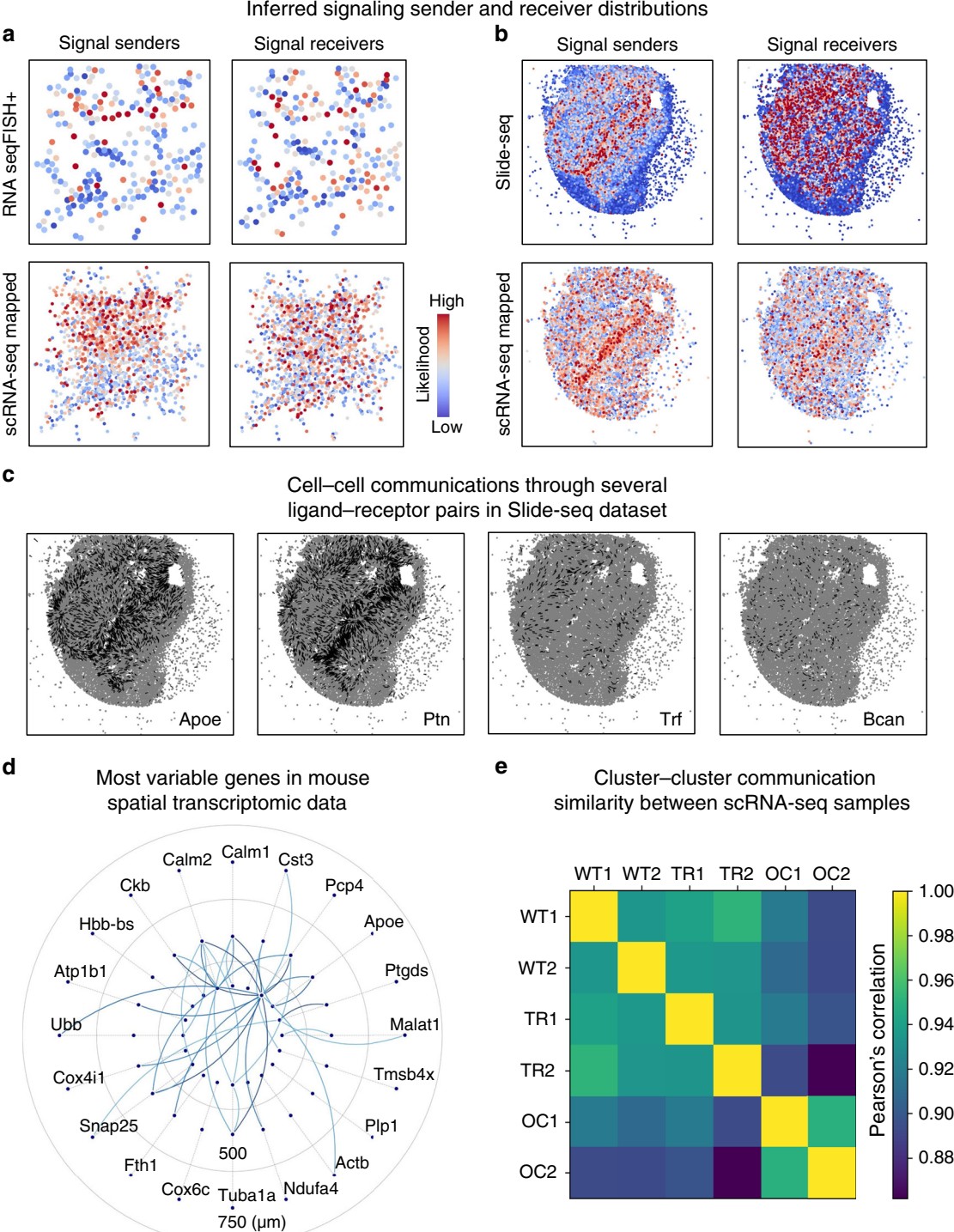

**Fig. 6 Application of SpaOTsc to spatial transcriptomic data and their integrations with scRNA-seq datasets.** The Slide-seq data, the RNA seqFISH+ data, and the scRNA-seq data for mouse olfactory bulb were taken from ref. [51], ref. [52], and ref. [53], respectively. **a, b** Spatial distributions of signal senders and receivers with the color showing the likelihood of being a sender or receiver. Top row: inference using only spatial transcriptomics data. Bottom row: the inferred signaling in scRNA-seq data (sample WT1) visualized by mapping the single cells to space using spatial transcriptomic data. **c** Signaling for four individual marked ligands in the Slide-seq data. **d** The intercellular gene–gene regulatory information flow of the top 20 variable genes in the Slide-seq data. **e** Similarities on cluster–cluster communication between the six samples of the scRNA-seq data integrated with the RNA seqFISH+ data measured by Pearson's correlation coefficient.

One can carry out three major tasks immediately after obtaining $\boldsymbol{\gamma}^*$ and $\widehat{\mathbf{D}}_{sc}$: (1) prediction of spatial gene expression at the $i$th position by $\sum_j \gamma^*_{j,i} \mathbf{g}_j / \sum_j \gamma^*_{j,i}$ where $\mathbf{g} \in \mathbb{R}^n$ is the expression vector for a gene in scRNA-seq data; (2) identification of spatially localized cell subclusters by distance-based clustering using $\widehat{\mathbf{D}}_{sc}$ within

each previously identified cluster; and (3) visualization of scRNA-seq data constrained by cell–cell distances using the distance matrix $\widehat{\mathbf{D}}_{sc}$.

The intercellular gene–gene regulatory information flow is inferred by using partial information decomposition[28,29,34]. We estimate how much unique information about a gene (target gene) can be provided by another gene (source

gene) in its spatial neighborhood through the calculation of the accumulated unique information:

$$u(G_{src}, G_{tar}, \eta) = \sum_{G \in \mathcal{G}} \text{Unq}_G(G_{tar}; \tilde{G}_{src}) \qquad (3)$$

where $G_{tar}$ is the variable for target gene expression in the cells, $\tilde{G}_{src}$ is the variable for source gene expression in $\eta$-neighborhoods of cells whose observation is estimated using $\hat{\mathbf{D}}_{sc}$, and $\mathcal{G}$ is a collection of genes with high intracellular correlation with the target gene. The unique information $\text{Unq}_X(Z;Y)$ measures how much unique information $Y$ provides about $Z$ in addition to $X$.

For the case of intercellular signaling with known ligands, receptors, and their downstream genes, we use random forest models[58,59] to infer the spatial distance of signaling. The ligand expressions of cells in a neighborhood of distance of $\eta$, denoted as $\tilde{L}(\eta)$, together with other genes highly correlated to a downstream target gene of the ligand–receptor interaction are used as features to fit a random forest model outputting the target gene. The receptor expressions are used as sample weights. The $\eta$ under which $\tilde{L}(\eta)$ has the highest feature importance is considered to be the spatial distance of this signaling.

Knowing the ligands, receptors and downstream genes involved in intercellular signaling and $\hat{\mathbf{D}}_{sc}$, we then infer cell–cell communication by solving another optimal transport problem. First, the source distribution over the cells $\boldsymbol{\omega}_L$ is constructed to be proportional to the expression of ligand gene. Next a destination distribution $\boldsymbol{\omega}_D$ is constructed based on the expression of receptors and downstream genes to represent the probability of a cell to receive the signal. A cell highly expressing receptors with downstream genes consistent with the up-/down-regulation relationships (low expression of down-regulated genes and high expression of up-regulated genes) is assigned with a high probability. With this information we solve the following optimal transport problem

$$\underset{\boldsymbol{\gamma} \in \mathbb{R}_+^{n \times n}}{\operatorname{argmin}} < \boldsymbol{\gamma}, \hat{\mathbf{D}}_{sc} >_F + \rho \big( \text{KL}(\boldsymbol{\gamma}\mathbf{1}^n | \boldsymbol{\omega}_L) + \text{KL}(\boldsymbol{\gamma}^T\mathbf{1}^n | \boldsymbol{\omega}_D) \big). \qquad (4)$$

The optimal transport plan $\boldsymbol{\gamma}_S^*$ is interpreted as likelihood of cell–cell communications, e.g. its $ij$th element describes how likely cell $j$ receives signal from cell $i$. When spatial distances for signaling are available, we can simply adjust the cost matrix $\hat{\mathbf{D}}_{sc}$ by setting entries greater than this distance to a large number to enforce a spatial constraint on communications identification. When a spatial constraint is applied, long-distance connections will be eliminated and new short connections may emerge (Supplementary Figs. 21, 22).

**Datasets and processing**. For zebrafish embryo, we downloaded the accompanying data files (https://www.dropbox.com/s/ev78jelev0jgu5s/seurat_files_zfin.zip?dl=1) for the Seurat tutorial (https://satijalab.org/seurat/seurat_spatial_tutorial_part1.html). The scRNA-seq data is stored in the file "zdata.matrix.txt" and the spatial data (in situ hybridization) is stored in "Spatial_ReferenceMap.xlsx"[13]. The scRNA-seq data is also available through the accession code GEO: GSE66688. We binarized the scRNA-seq data and selected a set of highly variable genes following the same tutorial. For the scRNA-seq data matrix $\mathbf{X}$, a log transformation was performed elementwise $\log(1 + \mathbf{X})$ for the analyses. Another more recent scRNA-seq data[36] (accession number: GSE112294) is used for the analysis of cell–cell communication. The cells for 6hpf are extracted followed by normalization to 10000 total counts per cell and a logp1 transform. Genes used for signaling analysis are listed in Supplementary Methods section 3.2.

For *Drosophila* embryo, the scRNA-seq data and the spatial data (in situ hybridization) were downloaded from the Dream Single cell Transcriptomics Challenge through Synapse ID (syn15665609)[10]. The files "bdtnp.txt" and "binarized_bdtnp.csv" were used for numerical and binary spatial data, respectively. The files "dge_normalized.txt" and "dge_binarized_distMap.csv" were used for the numerical and binary scRNA-seq data. The coordinate of each cell in the spatial data is assigned according to the file "geometry.txt". We used Scanpy[60] to select highly variable genes for downstream analysis (the script used is included in SpaOTsc tutorial files). Genes used for signaling analysis are listed in Supplementary Methods section 3.1.

For mouse visual cortex, the spatial data (Spatially-resolved Transcript Amplicon Readout mapping) was downloaded from STARmap Resources (https://www.dropbox.com/sh/f7ebheru1lbz91s/AABYSSjSTppBmVmWl2H4s_K-a?dl=0)[38]. We used the data named "20180505_BY3_1kgenes" from the folder "visual_1020". The scRNA-seq data was downloaded from Allen Brain Atlas[37,61] (http://celltypes.brain-map.org/api/v2/well_known_file_download/694413985), and specifically the file "mouse_VISp_2018-06-14_exon-matrix.csv" was used. The spatial data contains 1020 genes and quantifying similarity by directly computing correlation coefficients might include too much noise and inconsistency across datasets. Therefore, we used the "cca" utility in Seurat[15] which determines a low-dimensional common space for the two datasets and the script for processing is included in SpaOTsc tutorial files.

For mouse olfactory bulb, the spatial data by Slide-seq was downloaded from the Broad Institute Single Cell Portal (https://singlecell.broadinstitute.org/single_cell/study/SCP354/slide-seq-study)[51] with file ID: 180430_3. The spatial data by RNA seqFISH+ was downloaded from the Github repository (https://github.com/CaiGroup/seqFISH-PLUS)[52]. The scRNA-seq data were downloaded

from the supplementary of the associated publication[53]. The same procedure as for the mouse visual cortex data was used to measure similarities between cells of these two datasets. For the RNA seqFISH+ data with several fields, an initial mapping was done for each sample of the scRNA-seq data and the whole spatial data (all seven fields). The single cells were then assigned to the fields based on this initial mapping and separate mapping was carried out for each field. The 39 clusters of the scRNA-seq data were identified using the Scanpy package (PCA + Louvain)[60]. The ligand–receptor pairs were chosen from a ligand–receptor database[3] by using only secreted ligands according to the Human Protein Atlas[62]. The gene symbols were converted from human to mouse using the Mouse Genome Database[63].

**Reporting summary**. Further information on research design is available in the Nature Research Reporting Summary linked to this article.

## Data availability

The original data used in this paper can be accessed through the following links: (1) zebrafish embryo spatial data: downloaded from (https://www.dropbox.com/s/ev78jelev0jgu5s/seurat_files_zfin.zip?dl=1)[13]; (2) zebrafish embryo scRNA-seq data: GEO assession codes: GSE66688[13] (first dataset) and GSE112294[36] (second dataset); (3) *Drosophila* embryo spatial and scRNA-seq data: accessible at the Dream Single cell Transcriptomics Challenge through Synapse ID (syn15665609)[10]; (4) mouse visual cortex spatial data: downloaded from STARmap Resources[38] (https://www.dropbox.com/sh/f7ebheru1lbz91s/AABYSSjSTppBmVmWl2H4s_K-a?dl=0); (5) mouse visual cortex scRNA-seq data: downloaded from Allen Brain Atlas[37,61] (http://celltypes.brain-map.org/api/v2/well_known_file_download/694413985); (6) mouse olfactory bulb spatial data: Slide-seq data[51] downloaded from Broad Institute Single Cell Portal (https://singlecell.broadinstitute.org/single_cell/study/SCP354/slide-seq-study) and RNA seqFISH+ data[52] downloaded from (https://github.com/CaiGroup/seqFISH-PLUS); (7) mouse olfactory bulb scRNA-seq data: downloaded from the supplementary of the associated publication[53]. Full tutorials that reproduce the presented results containing the data used for analysis can be accessed through the GitHub repository (https://github.com/zcang/SpaOTsc).

## Code availability

An open-source Python implementation of SpaOTsc is available at GitHub (https://github.com/zcang/SpaOTsc).

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

## Acknowledgements

This work was supported by an NIH grant U01AR073159, an NSF grant DMS1763272, and a grant from the Simons Foundation (594598, QN).

## Author contributions

Z.C. and Q.N. conceived the method. Z.C. implemented the method and generated the results. Z.C. and Q.N. interpreted the results, generated the visualizations and wrote the paper.

## Competing interests

The authors declare no competing interests.
