## [Peer Review File · Nature Communications]

Reviewers' comments:

Reviewer #1 (Remarks to the Author):

The manuscript describes the application of a relaxed optimal transport algorithm to integrate scRNAseq with spatial imaging data to study cellular dynamics in a tissue context. The spatial tissue dynamics addressed in this manuscript include cell-cell communications and intercellular gene-gene regulatory relationships. A key step of the approach is the construction of a spatial metric with an optimal transport plan. The authors use partial information decomposition to understand indirect genes-genes interactions by quantifying the unique information provided by one gene to another gene across different cell types. Thorough application of the method for four pairs of scRNA-seq and spatial gene expression datasets was performed. Overall, this is a high-quality manuscript, reporting novel algorithms well implemented in a Python package. The manuscript was well-written with a clear description of concepts and algorithms, a comprehensive document describing the algorithms implemented in the SpaOTsc package is available and the software is well-documented. I believe the work has broad applications and will benefit the research community.

Some of my comments and questions are below:

The number of interactions detected using this method is very high. It should be more useful if the authors provide discussions and options for prioritising connections.

What are potential effects of different data normalisation and processing to the result of the SpaOTsc analysis pipeline?

How well the method can be applied for data from different experimental spatial technologies like Slide-seq, MERFISH, SeqFISH, which have different data characteristics? In this manuscript, the authors tested in situ hybridization data and STARmap data, but more recent and popular methods are Slide-seq, MERFISH, and SeqFISH.

In the SpaOTsc analysis reported in this manuscript, to assess the effect of the number of genes to the overall results, the author tested 10, 20 and 50 genes in Drosophila dataset. The authors claim that a moderate size of 50 genes is enough to capture the key features. However, this is a very small number of genes compared to the whole-genome coverage of over 32K genes in human or mouse genome. The authors may add discussions on the scalability of the methods when more genes and more cells are included. Preferably, a similar analysis to the Figs 15-17 but for the mouse visual cortex dataset should be added.

Supp Figure 11: Why with the additional spatial distance cutoff, new connections emerge? For example, comparing panel b to panel a, new connections between 4.1 and 5 in Wg signalling and 4.2 to 2.1 in the Dpp signalling.

Supp Figures 12 and 13: Plotting cells on tSNE to demonstrate effects of spatial distance cutoff to cell-cell connections is not a good way. The plots may have problems with reproducibility of the representation and potential confusion in interpreting tSNE distance and physical distance. tSNE plots do not accurately represent cell-cell distance in gene expression or physical distance space, for example most connections to cells in 1 in the tSNE plots are actually to cells in 7. How reproducible the many connections are displaced in the tSNE plots? If UMAP plots are used, are the connection patterns between clusters similar compared to tSNE plots?

What are colors in Supp Figures 11, 12, 13 represent? If the author use the colors to represent cluster labels, consistent colors and cluster number are required.

Please add X and Y axis label in the Supp Fig 15

Reviewer #2 (Remarks to the Author):

Cang and Nie introduce a computational method for integrating spatial and single-cell transcriptomic data based on optimal transport (OT) theory. With SpaOTsc, the authors were able to integrate spatial level data to dissociation-based single-cell RNA-seq data, in order to infer spatial distance between cells and use this information to drive dimension-reduced embedding and clustering of scRNA-seq data. They can also generate gene networks from spatial expression and identify clusters of genes that correspond to regions of the spatial transcriptomic data. Moreover, optimal transport is used to identify intercellular signaling based on known ligand, receptor, and downstream effector genes, and probabilities of signaling interactions constrained by spatial distance are produced. This work seems timely and exciting due to the types of data being generated in the field of single-cell biology, and would evoke substantial enthusiasm.

Given the enthusiasm, there is also some comments I would like to make about the paper. While SpaOTsc seems promising when benchmarked against DistMap, Achim, and Seurat V1, the utility of this tool for enhancing single-cell data using spatial transcriptomic methods has not been demonstrated. Primarily, it appears that the reconstructed spatial patterning of genes compared to experimental data only shows genes measured by the spatial methods themselves. I.e. the ability to predict spatial patterning of unmeasured genes was not sufficiently shown. This paper would be strengthened by an example of out-of-sample prediction of spatial expression of gene(s) not measured by FISH/STARmap using SpaOTsc mapping of scRNA-seq data. Results could be validated against known expression patterns within the sample/organism of interest (e.g. 6hpf zebrafish embryo expressing *admp* and *ved* at dorsal and ventral sites, respectively).

- Low-dimensional embeddings of scRNA-seq datasets using native cell information and inferred spatial cell-cell distance from SpaOTsc seem very similar. What is the functional difference between them and what does this tell you? If clustering and embeddings are nearly the same from the two types of data, does this suggest that spatial information is inherently present in the scRNA-seq data? Can you draw ventral/dorsal/anterior/posterior axes on a t-SNE/UMAP generated from spatial cell-cell distances? Moreover, there should be examples that the authors can use where space does not equal function (thinking of scenarios in the adult situation where cells for the most part are intermixed)
- How are gene sets for sender and receiver cells defined? Explanation of the reference database and/or the required information to run this signaling analysis de novo would be helpful.
- Links to code and tutorials do not work
- For the majority of the paper in both sets of data, the authors primarily focused on WNT and TGFB family signaling pathways. It leads the readers to think that this approach will only work for these pathways and not others (e.g. EGF signaling). It would be prudent to include examples of other pathway types that can be inferred
- I am not understanding the difference between Figure 4A and B. While 4A shows an inverse relationship regarding receiver and sender to be inversely correlated between the 2 pathways, Figure 4B shows a direct correlation. Why the discrepancy?

Response to Reviewer 1

General comments:

The manuscript describes the application of a relaxed optimal transport algorithm to integrate scRNAseq with spatial imaging data to study cellular dynamics in a tissue context. The spatial tissue dynamics addressed in this manuscript include cell-cell communications and intercellular gene-gene regulatory relationships. A key step of the approach is the construction of a spatial metric with an optimal transport plan. The authors use partial information decomposition to understand indirect genes-genes interactions by quantifying the unique information provided by one gene to another gene across different cell types. Thorough application of the method for four pairs of scRNA-seq and spatial gene expression datasets was performed. Overall, this is a high-quality manuscript, reporting novel algorithms well implemented in a Python package. The manuscript was well-written with a clear description of concepts and algorithms, a comprehensive document describing the algorithms implemented in the SpaOTsc package is available and the software is well-documented. I believe the work has broad applications and will benefit the research community.

Response: We thank the reviewer very much for the supportive comments and the following constructive specific comments.

Specific comments:

1. The number of interactions detected using this method is very high. It should be more useful if the authors provide discussions and options for prioritizing connections.

Response: We thank the reviewer for pointing this out and the nice suggestion. The number of interactions for Wg in the drosophila dataset was high while the number of

interactions for some other cases, such as the Wnt signaling in the zebrafish dataset, was relatively low in our study.

To follow the reviewer's suggestion of adding options for prioritizing connections, in the revision we have added a user-defined parameter to the package. When evaluating cell-cell communications, the significance of each gene involved was measured by a weight function (Supplementary Methods Section 1.6) based on the expression level (Supplementary Methods Eq. 34 and 35). One typical option of the weight function could be an exponential weight function $\phi(x) = \exp(-(x/\eta)^v)$. The scaling parameter η can be used to measure if a gene is more biased to the "off" state or the "on" state. In the revision, this parameter now has been changed as an input of the package instead of a given parameter in the original version. In the revision, we have systematically examined the effect of this parameter on the number of interactions (new Supplementary Fig. 16), and found that when increasing this parameter, a smaller number of essential "signal sender" cells and "signal receiver" cells were identified. The parameter η shows a nice control of the number of interactions according to the user's choices on the significance of the signal senders and signal receivers.

In the revision, we have added a few sentences to describe this parameter and the outcomes of tuning this parameter at the end of the fifth paragraph of the Results section "Reconstruction of cell-cell communications in space".

2. What are potential effects of different data normalization and processing to the result of the SpaOTsc analysis pipeline?

Response: Thanks for bringing up this good point. In the original publications containing the drosophila and zebrafish datasets, both the scRNA-seq and the spatial data were binarized first before being analyzed in the original study. We adapted the same approach in our original submission. In the revision we added studies of three other normalization procedures on the drosophila datasets including the default settings of Seurat and Monocle, and a normalized procedure used by the DREAM Single-Cell Transcriptomics

Challenge (Synapse ID: syn15665609), along with the original count matrix without any normalization.

Similar to the processing procedures for numerical data in the original submission (Fig. 2c), we used Spearman's correlation to quantify the similarity between single cells and spatial locations. Very similar performance was observed across all five different normalization procedures with average AUC's rounding up to 0.88 (Fig. 2b2 and new Supplementary Fig. 5).

Another key component is how to measure gene expression similarity between cells in scRNA-seq data and locations in spatial data. Since the scRNA-seq data and spatial data can have very different characteristics, we chose the measurements such as Matthew's correlation coefficient following the binarization of data (Fig. 2b2) and Spearman's correlation coefficient for numerical data. These measurements are usually robust to scales as opposed to some other commonly used ones such as Pearson's correlation coefficient. The robust prediction performance across different preprocessing procedures is likely due to these choices of similarity measurements. In the main figures, we used the processing steps of the original publications associated with the datasets because these processing procedures have shown proper characteristics of the corresponding biological systems. While we recommend using binarized data or ranking-based correlation coefficients on numerical data, we provide users with the flexibility to apply any of their own similarity measurements that they consider appropriate for the specific biological systems.

In the revision, we have added a few sentences at the end of the second paragraph of the Results section "Performance of SpaOTsc mapping and comparison to other methods" to discuss this point.

3. How well the method can be applied for data from different experimental spatial technologies like Slide-seq, MERFISH, SeqFISH, which have different data characteristics? In this manuscript, the authors tested in situ hybridization data and

STARmap data, but more recent and popular methods are Slide-seq, MERFISH, and SeqFISH.

Response: Thanks for bringing up this great point. In the revision we have added a study of two different spatial technologies for the mouse olfactory bulb (New Fig. 6, New Supplementary Fig. 30-35). In this analysis, we used two spatial datasets, a Slide-seq dataset (*Rodrigues, Samuel G., et al. Science 363.6434 (2019): 1463-1467*) and an RNA seqFISH+ dataset (*Eng, Chee-Huat Linus, et al. Nature 568.7751 (2019): 235*) to study the cell-cell communication. In addition we used a ligand-receptor database (Ramilowski, Jordan A., et al. Nature communications 6 (2015): 7866) containing more than a thousand ligand-receptor pairs to systematically identify communication links.

We carried out four signaling analyses: cell-cell communications based only on the two spatial datasets, and the cell-cell communications based on the scRNA-seq data with the two spatial datasets as the spatial reference. We found that the signal sending cells seem to have some spatial localization pattern while the signal receiving cells are relatively scattered over the space. The reconstructed spatial distributions of signal sending cells in the analysis of scRNA-seq data agree well with the signaling patterns based on only the spatial datasets.

The scRNA-seq data in this study contains six samples under three physiological conditions: wild type, olfactory trained, and naris occluded. We then explored if the cell-cell communication patterns under these different conditions are different. We found that while the wild type samples and the olfactory trained samples share similar communication patterns, the naris occluded samples show relatively different behavior.

Our method can be directly applied to the modern spatial transcriptomics datasets. The integration utility for scRNA-seq data and spatial data, however, can still be useful in some cases. For example, in these signaling analyses, there are 1157 ligand-receptor pairs present in the scRNA-seq data compared to 989 and 758 pairs in the Slide-seq dataset and the RNA seqFISH+ dataset. So scRNA-seq data may provide additional ligand-

receptor information. This can be especially useful if some key genes of interest are not measured in spatial data but are available in scRNA-seq data.

Moreover, one may encounter the situation for which scRNA-seq data of several samples are obtained under different physiological conditions. Using the integration utility of SpaOTsc, one can integrate the scRNA-seq data of each sample with a common spatial reference data annotating the locations of different cell states. Then, one will be able to study the difference in spatial activity between the scRNA-seq data of different samples. As an example, we studied the difference in spatial cell-cell communications across several olfactory bulbs under different physiological conditions using scRNA-seq data of several samples and one common spatial reference data.

In the revision, we have added a new section (“Applications to spatial transcriptomics datasets” in the Results section), a new main figure (New Fig. 6), along with six supplementary figures (New Supplementary Fig. 30-35) on the analysis of mouse olfactory bulb using scRNA-seq data and modern spatial transcriptomics datasets. We have also discussed the potential application of our method on modern spatial transcriptomic data in the Discussion section with new references added.

4. In the SpaOTsc analysis reported in this manuscript, to assess the effect of the number of genes to the overall results, the author tested 10, 20 and 50 genes in Drosophila dataset. The authors claim that a moderate size of 50 genes is enough to capture the key features. However, this is a very small number of genes compared to the whole-genome coverage of over 32K genes in human or mouse genome. The authors may add discussions on the scalability of the methods when more genes and more cells are included. Preferably, a similar analysis to the Figs 15-17 but for the mouse visual cortex dataset should be added.

Response: Thanks for the suggestions. In the revision we have systematically increased the number of background genes one at a time from 1 gene to 300 genes in the mouse visual cortex dataset containing 14249 high-quality cells (New Supplementary Fig. 28).

We stopped at 300 genes for this case because only about 300 highly variable genes were selected in the Scanpy package for preprocessing. We have also carried out a similar study from 1 gene to 500 genes for the drosophila embryo dataset (New Supplementary Fig. 27).

Pearson's correlation coefficients for both datasets between the connections identified using 50/50 background genes and using 300/500 background genes were found to be greater than 0.9. At a higher number of background genes, the identified connections stay almost identical (from 200 to 300 background genes in mouse visual cortex example and from 300 to 500 background genes in drosophila embryo example).

In the revision, we have added a paragraph at the end of the section "Identification of intercellular gene-gene regulatory information flows" to address this point.

5. Supp Figure 11: Why with the additional spatial distance cutoff, new connections emerge? For example, comparing panel b to panel a, new connections between 4.1 and 5 in Wg signaling and 4.2 to 2.1 in the Dpp signaling.

Response: Thanks for this great point. In our model, a signal sending cell has a fixed final amount "mass", which describes the level of gene expression. If one cell receives more signals from a sending cell, the other cells will receive less from the same sending cells. So removing one connection using distance cut-off will increase the available gene expression for other potential receiving cells. As a result, new connections become possible when removing one connection using distance cut-off. This property is reflected in our objective function for constructing the cell-cell communication network: 1) a penalty for long-distance transportation of ligand and 2) a penalty for violating the conservation of mass. The newly merged connections are consequences of the competition between these two components.

To make this point clear, in the revision we have added a simple example illustrating possible outcomes by adding a spatial constraint (New Supplementary Fig. 22). Besides

showing the changes in the top connections (Supplementary Fig. 18), we have added a New Supplementary Fig. 21 to show the changes in all the identified connections due to the addition of the spatial constraint in the form of heatmaps. This result has been added to the last paragraph of the Method section “SpaOTsc model” in the revision.

On the single-cell level, for illustration, we consider a simple system with three cells: a signal sender cell (cell 1) and two signal receiver cells (cell 2 and cell 3). Cell 3 is more likely a receiver cell than cell 2 (in other words, cell 3 has a greater mass than cell 2 as destinations in the optimal transport setup) but the distance between cell 3 and cell 1, $d(\text{cell 1, cell 3})$ is greater than $d(\text{cell 1, cell 2})$. Without the spatial constraint, we would identify a stronger connection between cell 1 and cell 3 than between cell 1 and cell 2 due to the conservation of mass. If a spatial constraint that is between $d(\text{cell 1, cell 3})$ and $d(\text{cell 1, cell 2})$ is applied, we would observe no mass going from cell 1 to cell 3 (thus the previous strong connection vanishes) but more mass going from cell 1 to cell 2 (the emergence of a shorter connection). In the optimal transport formulation, the transportation cost (a large cost if moving much mass along long distance) and the mass conservation violation (a penalty for unbalanced mass transport) are the two penalty terms in the objective function to be minimized. When a spatial constraint is applied, a connection whose distance is longer than this constraint will induce a huge penalty in the transportation cost while the elimination of this connection might lead to the emergence of new short connections which only induces a moderate penalty in mass conservation violation. Therefore, we may observe new connections when a spatial constraint is applied.

On the cluster level, there are also new connections of visually long distances when the spatial constraint is added (Supplementary Fig. 18). This is because we used a geometric average to represent the location of clusters. So there are clusters that are adjacent in space while their cluster centers are visually distant (e.g. cluster 4.1 and cluster 3.1 in Supplementary Fig. 18b). As a result, we may observe a new connection

between clusters of visually long distance contributed by cell pairs of short distance located near the interfaces where the two clusters meet.

6. Supp Figures 12 and 13: Plotting cells on tSNE to demonstrate effects of spatial distance cutoff to cell-cell connections is not a good way. The plots may have problems with reproducibility of the representation and potential confusion in interpreting tSNE distance and physical distance. tSNE plots do not accurately represent cell-cell distance in gene expression or physical distance space, for example most connections to cells in 1 in the tSNE plots are actually to cells in 7. How reproducible are the many connections displaced in the tSNE plots? If UMAP plots are used, are the connection patterns between clusters similar compared to tSNE plots?

Response: Thanks for this constructive suggestion. In the revision, we have changed all the tSNE visualization to UMAP visualization. The overall organization of cells is similar to the tSNE plots in the drosophila embryo dataset. The two visualizations are generally different in other datasets. Indeed, we found the new UMAP visualization generally better than the previous tSNE visualization in representing the global spatial arrangements. Especially for the mouse visual cortex dataset (New Supplementary Fig. 11), we used UMAP visualization while tSNE failed to reproduce the known global spatial organization.

Despite the general difference between the tSNE and UMAP visualizations, for a specific dataset (the drosophila embryo dataset), the connection patterns between clusters and the visualized cell-cell communications from the new UMAP visualization and the old tSNE visualization are very similar.

Regarding the question on the subset of cluster 7 cells located near cluster 1, we have found this spatial arrangement is still present in the new UMAP visualization. This is because the first level clustering (cluster 7 and cluster 1 are obtained from this first level clustering) was done with only single-cell RNA sequencing data. Though the drosophila

embryo dataset shows a strong correlation between space and function, there are cases where a cell cluster can have spatially localized subclusters. For example, this is the case for cluster 7 where a part of it is spatially close to cluster 1 and the other part is spatially close to cluster 6.

In the revision, we have added a sentence to the second paragraph of the Results section “Space-constrained visualization and clustering of cells and genes in scRNA-seq data” to clarify this point.

7. What are colors in Supp Figures 11, 12, 13 represent? If the author use the colors to represent cluster labels, consistent colors and cluster number are required.

Response: These colors represent the cluster/subcluster labels and are consistent with other plots on the drosophila embryo dataset in the paper. We have added descriptions to the figure captions.

8. Please add X and Y axis label in the Supp Fig 15

Response: We have added the X and Y axis labels.

Response to Reviewer 2

General comments:

Cang and Nie introduce a computational method for integrating spatial and single-cell transcriptomic data based on optimal transport (OT) theory. With SpaOTsc, the authors were able to integrate spatial level data to dissociation-based single-cell RNA-seq data, in order to infer spatial distance between cells and use this information to drive dimension-reduced embedding and clustering of scRNA-seq data. They can also generate gene networks from spatial expression and identify clusters of genes that correspond to regions of the spatial transcriptomic data. Moreover, optimal transport is used to identify intercellular signaling based on known ligand, receptor, and downstream effector genes, and probabilities of signaling interactions constrained by spatial distance are produced. This work seems timely and exciting due to the types of data being generated in the field of single-cell biology, and would evoke substantial enthusiasm.

Response: We appreciate the reviewer's positive comments and the following constructive specific comments.

Specific comments:

1. Given the enthusiasm, there is also some comments I would like to make about the paper. While SpaOTsc seems promising when benchmarked against DistMap, Achim, and Seurat V1, the utility of this tool for enhancing single-cell data using spatial transcriptomic methods has not been demonstrated. Primarily, it appears that the reconstructed spatial patterning of genes compared to experimental data only shows genes measured by the spatial methods themselves. I.e. the ability to predict spatial patterning of unmeasured genes was not sufficiently shown. This paper would be strengthened by an example of out-of-sample prediction of spatial expression of gene(s) not measured by FISH/STARmap using SpaOTsc mapping of scRNA-seq data. Results

could be validated against known expression patterns within the sample/organism of interest (e.g. 6hpf zebrafish embryo expressing *admp* and *ved* at dorsal and ventral sites, respectively).

Response: We apologize for the confusion. In the original submission, all the methods including ours were validated in the same Leave-one-out cross-validation setting, that is, part of the spatial data after subtracting the predicting gene was used for the prediction. This procedure provides a way of measuring the accuracy of predicting spatially unmeasured genes. We have added a detailed description to clarify the prediction procedure in the revision (second paragraph of “Performance of SpaOTsc mapping and comparison to other methods”).

Regarding the point that some additional blind prediction would be helpful where the test data is hidden from not only the model but also the researcher when developing the method, we gathered additional imaging data of 13 genes in drosophila embryo that were not in the spatial dataset in our original submission (new data taken from Supplementary Figure S4 of Karaikos, Niko, et al. *Science* 358.6360 (2017): 194-199.) showing various spatial patterns that are present in the scRNA-seq data but are absent in the spatial data. We observed that the predicted spatial patterns agree well with the added imaging data (New Supplementary Fig. 3). We have added this new result to the new Supplementary Figure 3.

2. Low-dimensional embeddings of scRNA-seq datasets using native cell information and inferred spatial cell-cell distance from SpaOTsc seem very similar. What is the functional difference between them and what does this tell you? If clustering and embeddings are nearly the same from the two types of data, does this suggest that spatial information is inherently present in the scRNA-seq data? Can you draw ventral/dorsal/anterior/posterior axes on a t-SNE/UMAP generated from spatial cell-cell distances? Moreover, there should be examples that the authors can use where space does not equal function (thinking of scenarios in the adult situation where cells for the

most part are intermixed)

Response: Thanks for the good questions.

For the datasets of early developing tissue (the drosophila embryo and the zebrafish embryo), the UMAP plot using the inferred spatial distance (spatial UMAP) and the UMAP plot using only the single-cell RNA sequencing data (scRNA-seq UMAP) are generally similar. However, the inferred spatial distance still provides some additional information. For example, the spatially localized subclusters, which are well separated in the spatial UMAP plot, are mixed in the scRNA-seq UMAP plot (e.g. cluster 7 has three spatially localized subclusters in Fig. 3b). Also, the relative distances between clusters in the spatial UMAP are more consistent with the geometric configuration (Supplementary Figs. 6,8). It seems even in the tissues where space corresponds strongly to function, our method still identifies spatially localized subclusters and renders reasonable relative spatial arrangements between cell clusters that are not observed in scRNA-seq only UMAP plot.

Regarding the point of tissue axes, the UMAP visualization using the cell-cell spatial distance can reproduce some spatial axes in the original tissue. For example, the ventral-dorsal/posterior-anterior axis for the drosophila embryo (Fig. 3b), the ventral/dorsal/animal sides for the zebrafish embryo (Supplementary Fig. 8), and the L2/3-L6 region axis for the mouse visual cortex (Fig. 3a).

In the revision, we also followed the reviewer's suggestion to test on a tissue where the cells of different types are partially intermixed for the mouse visual cortex system. The UMAP visualization based on the spatial distance shows that some cell types are intermixed, agreeing well with prior knowledge (Fig. 3a). On the other hand, the UMAP visualization based only on scRNA-seq data exhibits well-separated cell clusters of different types, however, it failed to reconstruct the known spatial layout from region L2/3 through L6 (New Supplementary Fig. 11).

In the revision, we have added a New Supplementary Fig. 11 to show the difference in visualization between using the inferred spatial distance and using only the scRNA-seq data on a mature tissue (the mouse visual cortex). In addition, we have changed all the visualization using tSNE in the original submission to UMAP for consistency and reproducibility. We have also added annotation of tissue axes on the UMAP visualizations based on the spatial distance (Fig. 3a,b, Supplementary Fig. 8).

3. How are gene sets for sender and receiver cells defined? Explanation of the reference database and/or the required information to run this signaling analysis de novo would be helpful.

Response: Thanks for the nice suggestion.

The signaling analysis of zebrafish embryo and drosophila embryo are based on the known ligand, receptor, and downstream genes extracted from publications and published databases. In the revision, we have added this information to the Data sets and processing section of the main paper. The genes used together with the corresponding references are listed in Supplementary Methods Section 3.

Due to the noisy behavior of scRNA-seq data, a single pair of ligand-receptor is often insufficient to depict the cell-cell communications, and inclusion of a few downstream genes in the ligand-receptor pathway is important to enable consistent inference. On the other hand, a large number of ligand-receptor pairs without including the downstream genes could be effective to infer the existence of cell-cell communication between cells or clusters. For example, in the revision, we added a ligand-receptor database (Ramilowski, Jordan A., et al. *Nature communications* 6 (2015): 7866.) of more than a thousand ligand-receptor pairs to infer the cell-cell communication in the added mouse olfactory bulb system.

In the revision, the references for all the genes used and the details of using the ligand-receptor database were added to Supplementary Methods. We have added a sentence

to paragraph 1 in the Results section “Overview of SpaOTsc method-Inference of cell-cell communication and intercellular gene-gene regulatory information flow constrained by spatial metric” to present the needed inputs for genes when analyzing intercellular signaling.

4. Links to code and tutorials do not work

Response: We apologize for the oversight and inconvenience. In the original submission, we used UCI Github and google drive for the link to code and tutorials. We recently noticed non-UCI users need special access permissions. In the revision, we now use regular Github, and the package is now available at github.com/zcang/SpaOTsc. We have made the full tutorial file available for everyone with the link that is included in the manuscript. We will dump the finalized version into a more permanent location once the revision process has finished.

5. For the majority of the paper in both sets of data, the authors primarily focused on WNT and TGFB family signaling pathways. It leads the readers to think that this approach will only work for these pathways and not others (e.g. EGF signaling). It would be prudent to include examples of other pathway types that can be inferred

Response: Thanks for this excellent suggestion. In the revision, we have added an FGF signaling analysis for the zebrafish embryo (New supplementary Fig. 14) and an EGF signaling for the drosophila embryo (New supplementary Fig. 23). These two signaling pathways are related to the signal analyses in the original publications that produced the two datasets. In our new study, FGF signaling was found to be abundant at the dorsal side of zebrafish embryo and BMP signaling was found to be more active in the ventral side, a result consistent with the literature suggesting that FGF downregulates BMP in early zebrafish embryo development (Fürthauer, Maximilian, et al. Development 131.12 (2004): 2853-2864). For the drosophila embryo, we analyzed EGF signaling which was known to regulate the dorsal-ventral patterning alongside Dpp signaling. We found that while Dpp signaling is more active at the anterior side, where Sog, a long-distance

transporter of Dpp, is highly expressed, the dorsal-ventral components of the inferred EGF signaling were found to be more active in the posterior.

In the revision, we have also added a study of the cell-cell communications in the mouse olfactory bulb (New Fig. 6, New Supplementary Fig. 30-35). In this example, we explored near a thousand ligand-receptor pairs across different signaling pathways. These ligand-receptor pairs are obtained by selecting secreted ligands from a published ligand-receptor database (Ramilowski, Jordan A., et al. *Nature communications* 6 (2015): 7866.).

6. I am not understanding the difference between Figure 4A and B. While 4A shows an inverse relationship regarding receiver and sender to be inversely correlated between the 2 pathways, Figure 4B shows a direct correlation. Why the discrepancy?

Response: Sorry for the confusion. The reviewer is correct that there is a common signaling receiver cell cluster between the Wnt signaling and the BMP signaling (cluster 9 located near the dorsal margin). For the BMP signaling, there are also other significant signal receiving clusters (located at the ventral side) in addition to cluster 9.

To make this point clearer, in the revision we have added histograms summarizing the total signal sending and receiving activities for each cluster in Fig. 4b to better show the significant signal receiving clusters located on the ventral side, along with the observed difference in the spatial distributions of signal receiving cells (The color map in New Fig. 4a).

REVIEWERS' COMMENTS:

Reviewer #1 (Remarks to the Author):

The authors have sufficiently addressed my comments.

Reviewer #2 (Remarks to the Author):

The authors have addressed my comments.